# An Experience Transfer Approach for the Initial Data of Iterative Learning Control

**Shaozhe Liu** [1], **Zuojun Liu** [1,*], **Jie Zhang** [2] **and Dong Hu** [1]

1   School of Artificial Intelligence, Hebei University of Technology, Tianjin 300131, China;
    liusz77@126.com (S.L.); hudongxy@163.com (D.H.)
2   School of Engineering, Merz Court, Newcastle University, Newcastle upon Tyne NE1 7RU, UK;
    jie.zhang@newcastle.ac.uk
*   Correspondence: liuzuojun@hebut.edu.cn; Tel.: +86-6020-3069

**Abstract:** Iterative learning control (ILC) requires that the operating conditions of the controlled system must remain unchanged in the repetitive learning process. If the parameters of system change, the former control experience of ILC would not be effective anymore. A new process of iterative learning has to restart, which will exhaust more time and resource. Compared with learning from zero experience, appropriate initial data for the first iteration could reduce the turns of iterations to achieve the target tracking accuracy. When the parameters of a linear system change, its structure and nature are still intrinsically related to the original system. So, if the experience obtained from original ILC could be correspondingly adjusted according to the difference of new and original system, and use the adjusted experience as the initial data in the new iterative learning process, it would reduce the time and save the resource in the new ILC. Based on the idea of experience inheritance and transform, an experience transfer approach for the initial data of ILC is proposed in reference to the relation between the new and original systems. In this paper, via the method of recombining, translational and amplitude adjusting, the experience of former ILC is transferred as the initial control data of new ILC. Simulation shows that the convergence iteration of ILC with experience transfer approach reduces 55–75%, which demonstrates the effectiveness and advantages of the approach proposed in this paper. Both the deviation of the first iteration in ILC and the turns of iterations for achieving desired accuracy are reduced greatly.

**Keywords:** iterative learning control (ILC); experience inheritance; experience transform; initial iterative control data

## 1. Introduction

Iterative learning control (ILC) is a control method that is suitable for the repetitive control processes, such as reciprocating robot manipulator and multi-batch chemical production processes. In each of these tasks, the system is required to perform the same task over and over again. The learning process uses information from previous repetitions to improve the control data for the current repetition. It can efficiently track a desired trajectory for an uncertainty nonlinear dynamic system without requiring the mathematics model of the system. So, ILC is a model-free data-driven control method. Additionally, it has important practical value for high-precision industrial control systems without an accurate mathematic model [1,2]. After more than 40 years of development, ILC have made many important achievements in theoretical research [3–5] and practical applications [6–8].

The premise of ILC is that the parameters, desired trajectory and initial state of the system must remain unchanged for all repetitive periods [9]. However, if the premise cannot be strictly met, methods should be discovered to expand the scope of application of the ILC, or continue to use the control data obtained from the previous iterative learning, and avoid relearning from the beginning; this is the focus of many scholars in this field.

For the situations in which the desired trajectory is not strictly consistent or there is an initial state deviation, Hou proposed to use an extended state observer to estimate the non-strict repetitive disturbance and cancel its influence in advance. It made ILC applicable to the system with limited disturbance [10]. Sun proposed an iterative learning algorithm with initial correction effect and final state attraction, which effectively solved the problem of trajectory deviation in the case of fixed deviation between the iterative initial state and the desired initial state [11]. Liu proposed a flexible ILC for systems with similar desired trajectories by means of Cartesian product [12]. Foudeh proposed gradient-based norm optimal ILC for unmanned aerial vehicles to cope with exogenous disturbances caused by wind gusts [13]. Chen developed a distributed controller to solve the leader-follower consensus of multiple flexible manipulators with uncertain parameters, unknown disturbances, and actuator dead zones, in which iterative learning is used to handle the repeatable disturbances [14]. All the above studies mainly focus on tracking the trajectory with certain deviation or disturbance. There is no pre-adjustment of the ILC experience data according to the deviation or disturbance.

In iterative learning control, the initial iteration control data are usually set to zero or a constant. Once the controlled system or the desired trajectory changes, ILC needs to relearn from the beginning. As a result, in many high-precision machining processes, the original ILC experience is no longer applicable in the situations like a component is replaced, or the working environment changes, or the desired trajectory is adjusted. So ILC needs to go through many turns of learning again. In this process, a large amount of waste will be generated, leading to serious loss of time, materials and energy. Therefore, the research of inheriting the learning experience of former ILC in the new working conditions becomes a very necessary problem.

Some researches tried to inherit the experience of former ILC as the initial data of new ILC process when the desired trajectory changes. Xu designed a direct learning control method according to the proportional relationship between the new desired trajectory and the former desired trajectory in the amplitude axis and time axis, respectively [15,16]. After transforming the former ILC control data on the amplitude axis and time axis, it is used as the initial iterative control data of the new desired trajectory. Pu divided the gradient surface of the machined parts into a group of homogeneous trajectory groups [17], with final ILC control data of the first trajectory acting as reference. Then, those ILC control data are transformed via proportional scale and offset, which are used as the initial data of the next adjacent trajectory iterative learning. Hoelzle expressed the new desired trajectory as combinations of serial former trajectories, and combined the control data obtained from the former ILC as the initial iteration data of the new desired trajectory [18]. Xu proposed an ILC algorithm for optimal matching of trajectories [19]. The new desired trajectory is segmented, translated and rotated to obtain the primitive combined trajectory. Through the series combination of trajectory, control data and time-scale transformation, the initial iteration control data are obtained. Pang used the control data of symmetric trajectories for iterative learning control via mirror transformation [20]. Alajmi constructed an initial iterative control data set containing multi-frequency components of the system [21]. He also introduced an upper limit to the initial data for the input signal of ILC to avoid the aggressive response due to the uncertainty that lies in high frequencies. Besides, the robust iterative learning model predictive control algorithm with variable reference trajectory proposed by Ma also aimed at solving the control experience inheritance problem of ILC when the desired trajectory changes [22]. These studies used the pre-adjusted experience data from former ILC for a new desired trajectory. As the convergence of the error depends highly on the initial choice of input applied to the ILC, a set of good initial data would make learning faster and, as a consequence, the error tends to the required accuracy faster as well.

All the above studies only consider the ILC experience inheritance in the situations where the desired trajectory changes in the new control task. However, there are some other neglected situations which need ILC experience inheritance. For example, a new motor with higher power substitutes the old one in the robot manipulator driving unit, or some other parameters like the load, the damp ratio changes in the system. Until now, no research involving the ILC experience inheritance in such situations has been reported.

In modern industry, the changes of working process occur frequently. As the structure and essence of the system remain unchanged, so it is possible to transfer the former control experience or knowledge into the new system. For such systems, according to the specific relation within the parameters change, the inner relevance of two systems, the experience obtained from the former ILC might be inherited and transformed as the initial iterative control data of the new iterative learning process, which could save the time and resource in the new system.

The paper is organized as follows. In the second section, the traditional ILC algorithm is introduced. Its limitations are described. In the third section, an ILC of recombining transform strategy is presented. In the fourth section, an ILC of translational adjusting strategy is introduced. In the fifth section, an ILC of amplitude adjusting strategy is introduced. The simulation results are given to illustrate its merits over the traditional ILC in each section. Next, the convergence property of the ILC with experience transfer is analyzed. The conclusion is drawn in the final section.

## 2. Problem Description

### 2.1. Iterative Learning Control

ILC aims at a specific controlled system with a repetitive working process. ILC uses its historical operation data to modify the control action continuously, so as to improve the control accuracy. The typical P-ILC algorithm is described as follows:

$$
\begin{aligned}
u_{i+1}(k) &= u_i(k) + L * e_i(k) \\
e_i(k) &= y_d(k) - y_i(k)
\end{aligned}
\tag{1}
$$

where $i$ is the index of iterations; $k$ is the time of discretization; $u_{i+1}$ is the control input of the $(i+1)$-th iteration; $u_i$ is the control input of the $i$-th iteration; $L$ is the learning gain and $e_i$ is the error between the desired $y_d$ and the output $y_i$ in the $i$-th iteration process.

It can be seen from the above that the control input $u$ is adjusted iteratively on the iteration axis according to the input and output error of former iterations, which can be written as follows:

$$
u_{i+1}(k) = u_0(k) + L * \sum e_i(k),
\tag{2}
$$

where $u_0(k)$ is the initial control input of iterative learning. If the initial control input of the first iteration could be close to the control input of the final iteration, the turns of iterations required to achieve the desired control accuracy will be significantly reduced.

### 2.2. Description of Controlled System

The transfer function and state equation of the controlled system are generally described as following a transfer function or state equation:

$$
G(s) = \frac{K}{\prod (Ts + 1)} \cdot e^{-\tau s},
\tag{3}
$$

$$
\begin{aligned}
\dot{X} &= AX + BU(t - \tau) \\
Y &= CX
\end{aligned}
\tag{4}
$$

In the transfer function shown in Equation (3), $K$ is the system gain, which is generally affected by the driving power and load. It is related to the $B$ in the state Equation (4). The $T$ in transfer function is the inertia time constant of controlled system, which is related to the parameters of $A$ in the state equation. $\tau$ is the delay of control system, which is related to

the action process of actuator. The matrix *C* in the state equation is the parameters related with system output.

In the application of ILC, the parameters in Equation (3) or (4) must be the same in all iterations. However, sometimes, one of the parameters in the system might change due to the replacement of a unit in the system, such as a motor, a sensor, a harder material, a lighter load, etc. Although one parameter changes in the system, the structure and the order of the transfer function or state equation remain unchanged. This paper focuses on the above situations and manages to transfer the experience obtained in the ILC of the former system into the new system. In this paper, only the following changes are considered: the change of transfer function on time constant *T* of dominant pole due to the change of controlled system's damping, the change of state equation's parameters in matrix *A* due to parts' replacement, the change of parameters in the matrix *B* due to the change of power intensity in the driving unit, the change of the parameters in the matrix *C* due to change of the sensor's proportional coefficient in the sampling unit, and the change of the lag time in the system input and feedback units due to the replacement of actuators or sensors.

### 2.3. Problems Raised

In the industrial production or process control, when parts of the equipment or the materials of the processed parts are changed, the experience gained from original iterative learning control is no longer applicable. In general, learning should be resumed from zero initial data. In this re-learning process, there would be a negative impact on production efficiency and cost. For the mass production, this negative impact can be ignored, considering that tens of thousands of products are produced. However, for he small batch production, it is intolerable to ignore the negative impact caused by the continual changing of production process and equipment, which poses new challenges to the application of traditional ILC. How to effectively use the original control experience data of ILC in the new tasks becomes an important issue.

If the existing iterative learning control experience of the original system is directly taken as the initial iterative learning control value of the new system, the effect is generally better than that of learning from zero initial data. However, it is just a simple direct inheritance of experience. It lacks the necessary pre-adjustment according to the specific differences between the old and new systems. So, there is still room for further optimization in the setting of the initial iterative control data according to the experience obtained from the former ILC.

In general, when the parameter in the system is changed, e.g., the power of driving motor, the damping of the pipeline, system lag, and so on, the difference between the old and the new system can be checked via the equipment manual or be measured through tests. This is similar to the situation of an experienced technician who is asked to deal with a new material. The technician can compare the performance difference of new material with the former ones via testing or reading the users' manual of the new material. Then the difference could be used in the adjustment of the working experience. As a result, the technician could soon be able to make qualified products with the new material. In this paper, this process is called iterative learning control with experience transfer.

In this paper, the initial data of ILC are studied. The experience gained from the ILC in the original system is inherited and transformed to decrease the iteration turns in the ILC of new system, as shown in Figure 1. As the method proposed in this paper only changes the initial data of ILC, there is no change to the algorithm itself. So, the convergence property of the ILC with experience inheritance and transform is not changed too.

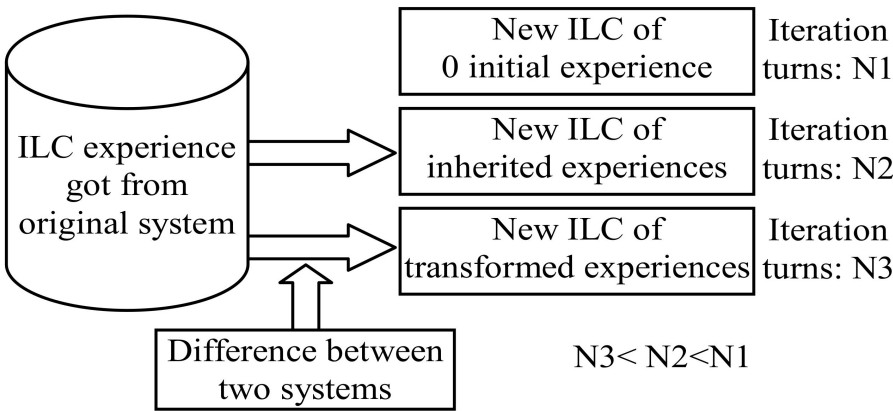

**Figure 1.** Iterative Learning Control (ILC) with Different Initial Experiences.

### 3. ILC of Recombining Transform

*3.1. Problem Description and Process Analysis*

In the mathematic model of control systems, when the damping of the system is changed, the inertial time parameters in the transfer function and the matrix *A* of the state equation are usually changed. When ILC is applied to this kind of system again, an ILC of re-combining transform strategy is proposed based on the tests of step response and impulse response. The re-combined experience is transformed as the control data of the initial iteration learning in the new system.

Firstly, the inertia time constant of the original and new systems are estimated by open-loop test of unit step response, as shown in Figure 2.

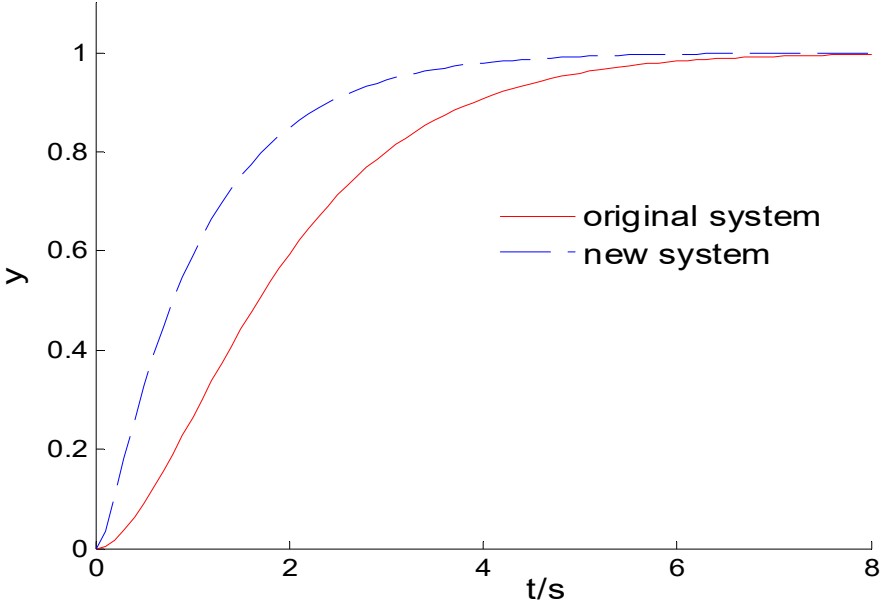

**Figure 2.** Step Response of Systems with Different Time Constants.

Secondly, according to the slope of the system's step response curve and damping, the two systems are modeled approximately. The unit impulse responses of two systems are simulated and analyzed respectively, as shown in Figure 3.

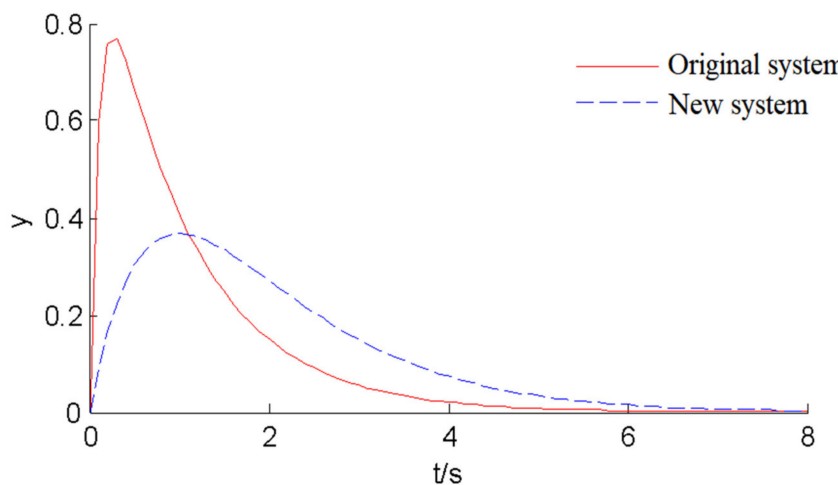

**Figure 3.** Impulse Response of Two Systems.

Next, record the impulse response of the new and the original controlled systems as sequence $y_a$ and $y_b$, respectively. The experience data sequence of ILC is regarded as a set of impulse functions that are input into the controlled system in a timely and sequential way, and the following quantitative analysis is carried out.

In the original controlled system, the inertia time constant is set as $T_a$. Then, the control value sequence obtained after $i$ turns of iterative learning is $U_{a,i}$. The control value $u_{a,i}(k)$ at a certain time $k$ on the corresponding time axis can be expressed as:

$$
\begin{aligned}
u_{a,i}(0) &\Leftrightarrow y_{a,i}(0,1)z^{-1} + y_{a,i}(0,2)z^{-2} + \cdots + y_{a,i}(0,p)z^{-p} \\
u_{a,i}(1) &\Leftrightarrow y_{a,i}(1,2)z^{-2} + y_{a,i}(1,3)z^{-3} + \cdots + y_{a,i}(1,p+1)z^{-p-1} \\
&\quad\quad\quad\quad\quad \cdots \\
u_{a,i}(k) &\Leftrightarrow y_{a,i}(k,k+1)z^{-k-1} + y_{a,i}(k,k+2)z^{-k-2} + \cdots + y_{a,i}(k,k+p)z^{-k-p}
\end{aligned}
\tag{5}
$$

where $p$ is 3 times of the system's inertia time constant $T_a$. $z^{-i}$ denotes the delayed effect of pulse input in the discrete system. The subsequent effect after $p$ could be considered to decay towards 0 and be ignored.

Similarly, in the new controlled system, the inertia time constant is $T_b$. Its initial iterative learning data is supposed as $U_{b,0}$. The control value sequence $u_{b,0}(k)$ at a certain time $k$ on the corresponding time axis can be expressed as:

$$
\begin{aligned}
u_{b,0}(0) &\Leftrightarrow y_{b,0}(0,1)z^{-1} + y_{b,0}(0,2)z^{-2} + \cdots + y_{b,0}(0,q)z^{-q} \\
u_{b,0}(1) &\Leftrightarrow y_{b,0}(1,2)z^{-2} + y_{b,0}(1,3)z^{-3} + \cdots + y_{b,0}(1,q+1)z^{-q-1} \\
&\quad\quad\quad\quad\quad \cdots \\
u_{b,0}(k) &\Leftrightarrow y_{b,0}(k,k+1)z^{-k-1} + y_{b,0}(k,k+2)z^{-k-2} + \cdots + y_{b,0}(k,k+q)z^{-k-q}
\end{aligned}
\tag{6}
$$

where $q$ is 3 times of the system's inertia time constant $T_b$. Additionally, the subsequent effect after $p$ could be considered to decay towards 0 and be ignored.

In Equation (5), the output response sequences $y_a$ of control data sequence $u_a$ combine together to form the whole desired trajectory. So do those of $y_b$. The control data sequences in Equation (6) are desired to have the same effect as that of Equation (5). The relation can be obtained via the discrete impulse response of the two systems shown in Figure 3. According to the energy-result balance and superposition principle in the linear system, there is the following relation:

$$u_{b,0}(0) = u_{a,i}(0)\frac{y_a(1)}{y_b(1)}$$

$$u_{b,0}(1) = u_{a,i}(1)\frac{y_d(2) - u_{a,i}(0)y_a(2)}{y_d(2) - u_{b,0}(0)y_b(2)} \cdot \frac{y_a(1)}{y_b(1)}$$

$$u_{b,0}(2) = u_{a,i}(2)\frac{y_d(3) - u_{a,i}(0)y_a(3) - u_{a,i}(1)y_a(2)}{y_d(3) - u_{b,0}(0)y_b(3) - u_{b,0}(1)y_b(2)} \cdot \frac{y_a(1)}{y_b(1)} \tag{7}$$

$$\cdots$$

$$u_{b,0}(k) = u_{a,i}(k)\frac{y_d(k) - \sum\limits_{j=0}^{k-1} u_{a,i}(j)y_a(k-j+1)}{y_d(k) - \sum\limits_{j=0}^{k-1} u_{b,0}(j)y_b(k-j+1)} \cdot \frac{y_a(1)}{y_b(1)}$$

So, using the discrete $y_a$ and $y_b$ sequence in the impulse response simulation, the $u_{b,0}(k)$ sequence can be determined. In this way, the control data sequence of the initial iterative learning data in the new controlled system can be obtained.

### 3.2. Simulation Test

For the approach described above, this paper uses two systems, as shown in Equations (8) and (9), to track the desired trajectory, as shown in Equation (10), for the simulation test.

$$G_a(s) = \frac{1}{s^2 + 2s + 1}, \tag{8}$$

$$G_b(s) = \frac{1}{0.1s^2 + 1.1s + 1}, \tag{9}$$

$$y(t) = 0.006t^2(20 - t) \quad t \in [0, 20] \tag{10}$$

Take the example of an original system $G_a$, as shown in Equation (8). Suppose the inertia time constant changes due to the replacement of a unit in the system, and the transfer function of system changes as shown in Equation (9). Via the relation of unit step response, the inertia time constant $T$ of two systems could be measured. Moreover, the unit pulse response of two systems can be figured via mathematic simulation. The control experience data obtained via iterative learning in system $G_a$ are inherited and transformed according to the methods in Equations (5)–(7). Then the experience data are transferred as the data of initial iterative learning data in $G_b$.

The simulation results are shown in Table 1 and Figure 4. The top figure shows the iterative learning process of the original system on the condition of zero initial data. The middle figure shows the iterative learning process of the new system when the initial data are obtained. The dashed line in the bottom figure shows the error convergence process based on ILC of the experience transform approach proposed in this paper. The solid line is the error convergence process of ILC without experience transform. According to the comparison results in the figure, the approach based on ILC of experience transform is better than that based on zero initial data in terms of initial deviation and iteration turns required to achieve the same control accuracy, e.g., 0.1 drawn by a dotted line in the figure of RMS error, which are 9 and 20, respectively. The learning efficiency improves by 55%.

**Table 1.** Effect of Recombined Experience Transform.

|                 | Initial Error | Convergence Iteration |
| --------------- | ------------- | --------------------- |
| Original system | 4.7           | 20                    |
| New system      | 1.0           | 9                     |

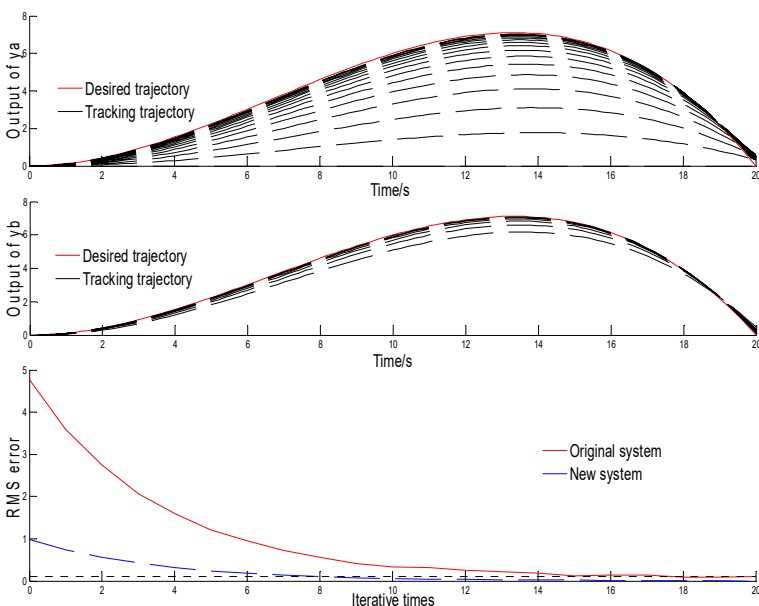

**Figure 4.** ILC of Recombined Experience Transform.

## 4. ILC of Translational Adjusting

### 4.1. Problem Description and Analysis

Aiming at a situation in which the delay time of control system changes, such as the increase of a conveying pipe's length, the decrease of a heating device's preheating time or the change of the feedback delay, and so on, an ILC of translational transform strategy is proposed according to the change of delay time. The translational adjusted experience is used as the control data of the initial iteration learning in the new system.

Take the chemical process shown in Figure 5 as an example. In the original system, valve *A* is used to control the input of the reaction tank, and the inlet flow is adjusted continuously with time according to the process requirements. Suppose that there is a fault in valve *A*, and valve *B* is used as backup. The delay time of pipeline transportation is increased by $\tau$, corresponding to *l* sampling periods. Then the ILC of translational adjusting equation for the new process is as follows:

$$\begin{aligned} u_{b,0}(k-l) &= u_{a,i}(k) \\ u_{b,j+1}(k) &= u_{b,j}(k) + L * e_{b,j}(k+l) \end{aligned}$$ (11)

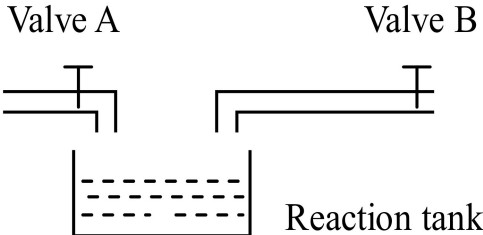

Valve A          Valve B

Reaction tank

**Figure 5.** System with Different Driving Units.

In Equation (11), for the increase of the delay time in the new system, it is necessary to shift the original control experience sequence $U_{a,i}$ along the time axis for *l* sampling periods to obtain the control data sequence $U_{b,0}$ of the new controlled system for the initial iterative learning.

### 4.2. Simulation Test

For translational experience transform, the system shown in Figure 5 is used for simulation test. The process of valve *A* is taken as the original system, and the proportional ILC is used to obtain the control experience. The process of valve *B* is taken as the new system. The control experience data obtained in system *A* are inherited and transformed as the control data of initial iteration in the way shown in Equation (8).

The process of experience transfer needs the relation/difference information of the original system and new system. The experience of former ILC will be transformed according to this information. To prove its effect in the situation of inaccurate measurement of delay time, a random measurement error is added to both the original system and the new system. The smaller measurement error, the smaller initial error and more reduced iteration time in new ILC will be obtained.

The simulation results are shown in Table 2 and Figure 6. It can be seen that the ILC with experience transfer is better than the ILC without experience transfer in terms of the initial deviation and the iteration turns required to achieve the same control accuracy, say 0.1, which are 6 and 20, respectively. The learning efficiency improves by 65%. For the case of shorter delay time, it is necessary to make forward translation in a similar way, and the proposed approach is also applicable.

**Table 2.** Effect of Translational Experience Transform.

|  | Initial Error | Convergence Iteration |
|---|---|---|
| Original system | 4.7 | 20 |
| New system | 0.6 | 7 |

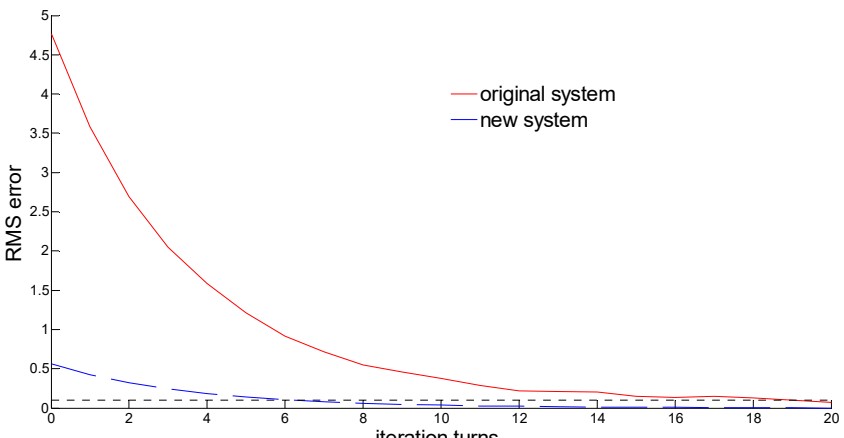

**Figure 6.** ILC of Translational Experience Transform.

## 5. ILC of Amplitude Adjusting

### 5.1. Problem Description and Process Analysis

Aiming at the situation that the driving power of the controlled system changes, such as the replacement of high-power electric heating elements or motors, the change of the conveying pipe's cross-sectional area, or the improvement of the output efficiency, an ILC of amplitude transform strategy is proposed based on the change of driving power. The amplitude-adjusted experience is used as the control data of the initial iteration learning in the new system.

First, the open-loop tests of unit step response are made to estimate the difference of the response amplitude between the old and the new controlled systems. The unit step response obtained is shown in Figure 7.

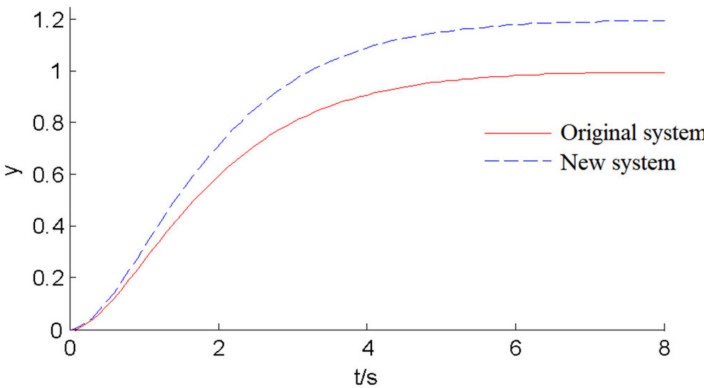

**Figure 7.** Amplitude Difference of Step Response.

Then the differences of driving power between the new and old processes are analyzed. The initial iterative learning data sequence $U_{b,0}$ of the new system is obtained by ILC of amplitude adjusting. If the change in step response is caused by friction or energy loss, the method of longitudinal shift amplitude adjusting as in Equation (12) is adopted. If the change of step response is caused by the change of total input power, the method of the proportional amplitude adjusting as in Equation (13) is adopted.

$$u_{b,0}(k) = (1 - \frac{y_b - y_a}{y_a})u_{a,i}(k), \tag{12}$$

$$u_{b,0}(k) = \frac{y_b}{y_a}u_{a,i}(k), \tag{13}$$

Finally, the conventional iterative learning control can be applied to carry out a new iterative learning process based on the initial data of $U_{b,0}$.

If the parameter in matrix $C$ of the state equation changes, the initial control data in the new iterative learning can also be obtained according to Equation (13).

### 5.2. Simulation Test

For the experience transform of longitudinal shift amplitude adjusting, a control system of robot manipulator is taken as example. Suppose that the moment of inertia ratio between the load and robot arms is 0.8:0.2. The old arm is replaced by a light one with the 50% moment of inertia, and the load and the driving motor remain unchanged. The ILC with experience transform pre-set its initial iteration learning data of the new system according to Equation (12). In addition, a certain random measurement error is taken into account in the simulation.

For the experience transform of proportional variable amplitude adjusting, take the similar example of robot manipulator. In this example, the driving power of the manipulator is supposed to increase 20%. The ILC experience transformation of the original system is carried according to Equation (13). The simulation results of the two kinds of ILC with amplitude adjusting are shown in Table 3 and Figure 8.

**Table 3.** Effect of Experience Amplitude Adjusting Transform.

|  | Initial Error | Convergence Iteration |
|---|---|---|
| Original system | 4.7 | 20 |
| Longitudinal shift new system | 0.5 | 7 |
| Proportional new system | 0.4 | 5 |

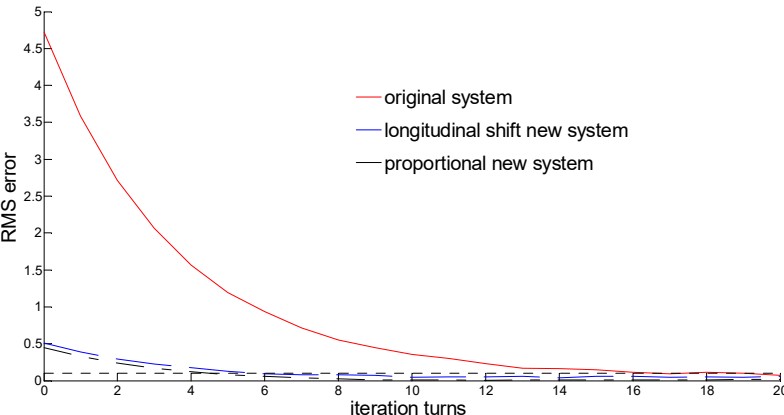

**Figure 8.** ILC of Experience Amplitude Adjusting Transform.

According to the comparison results in the figure, the iteration turns required for the same accuracy, say 0.1, are 7, 5 and 20, respectively. The learning efficiency improves 65% and 75%. The approach based on ILC of experience transform shows better initial deviation and convergence turns.

## 6. Convergence Analyze

The general convergence condition of ILC is shown below

$$\|I - CBL\| \leq 1, \tag{14}$$

where *I* is identity matrix; *L* is the learning gain of ILC; *C* and *B* are matrix in the system state equation. In this paper, if the parameters in *C* and *B* change, the *L* should be regulated too. As mentioned in Section 5.1, to replace the electric heating elements or motors with another one of different rated power, the change of the conveying pipe's cross-sectional area, or the improvement of the output efficiency, would cause the change of *B*. The change of sensors or amplifiers in sampling circuit would cause the change of *C*. To guarantee the convergence and the speed, the learning gain *L* should be regulated accordingly.

The ILC with experience transfer from former ILC only changes the initial data in new ILC process. It does not change the nature of the algorithm itself; neither the learning gain, nor the algorithm mechanism. So, the convergence property of the ILC with experience inheritance and transform can be guaranteed.

As the initial data in new ILC process are transferred from former ILC process with the reference information according to the difference between the new system and the former ones, the system could be accelerated to reach the required convergent accuracy with fewer error reduction iteration turns, which has been supported by the simulation tests in Sections 3–5. As a result, the efficiency of ILC for the new system could be improved.

## 7. Conclusions

For the linear system with fixed structure and essential characteristics, when the inertia time, lag time and driving power of the system change, this paper uses the approach of recombining, translational and amplitude adjusting to form the ideal initial iteration learning data for the new ILC process. The ILC with experience transfer reduces the iterations turns of iterative learning in the new process. Through a simulation test, the feasibility and effectiveness of the approach are demonstrated. It works well in the system with changed inertia time, lag time and driving power. Although the type of ILC algorithm used in the simulation is simple structure P in discrete-time, there is no limitation for the other type of ILC. The approach of experience transfer proposed in this paper is also effective for

PD(proportional-differential)-ILC, D(differential)-ILC or other ILC. Take a PD-ILC and a new controlled system as example:

$$u_{i+1}(k) = u_i(k) + 0.8e_i(k) + 0.1[e_i(k) - e_{i-1}(k)] \tag{15}$$

$$A = [-5\ 0; 0\ -6];\ B = [1; 1];\ C = [0\ 1];\ D = 0; \tag{16}$$

The desired trajectory is a sine curve. Suppose the parameter in the new system changes by 20%. The simulated results of experience transfer are shown in Table 4 and Figures 9–11.

**Table 4.** Effect of Experience Transfer for PD-ILC.

|  | Initial Error | Convergence Iteration |
|---|---|---|
| Original system | 0.72 | 8 |
| Recombined new system | 0.25 | 5 |
| Translational new system | 0.21 | 3 |
| Longitudinal shift new system | 0.15 | 3 |
| Proportional new system | 0.13 | 2 |

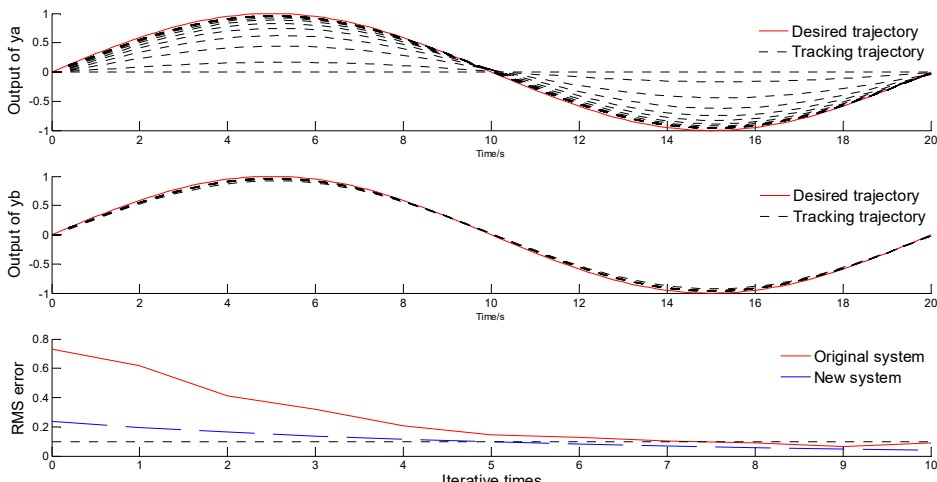

**Figure 9.** PD-ILC/Sine Curve of Recombined Experience Transform.

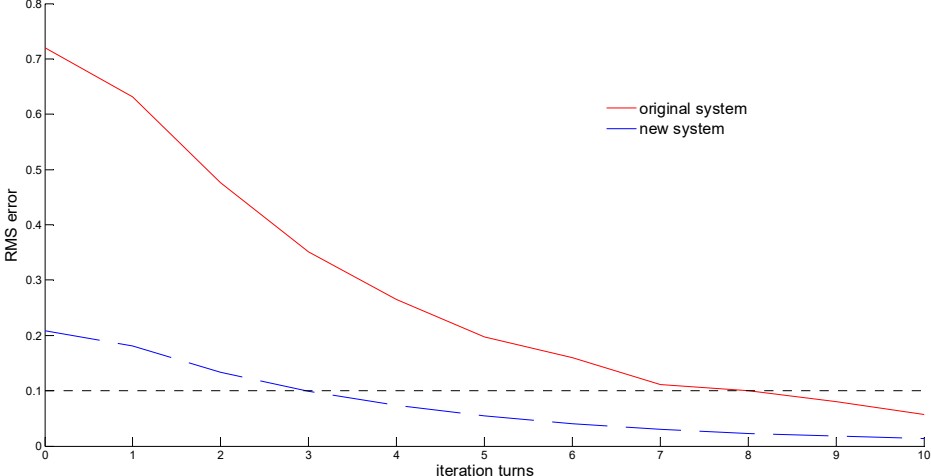

**Figure 10.** PD-ILC/Sine Curve of Translational Experience Transform.

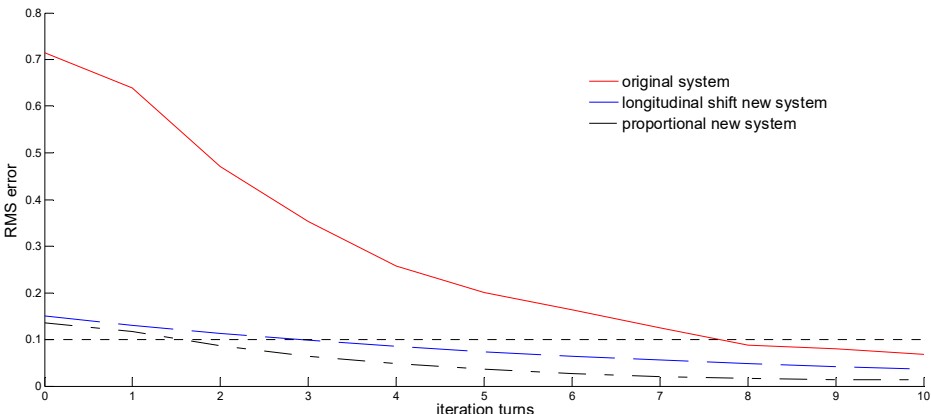

**Figure 11.** PD-ILC/Sine Curve of Experience Amplitude Adjusting Transform.

As shown in the examples above, the feasibility of the experience transfer approach for ILC is proved. In future work, it might be trialed in the nonlinear system, in which the nonlinear features would be also considered in the process of experience transfer.

**Author Contributions:** Investigation, D.H.; Methodology, Z.L.; Software, S.L.; Validation, Z.L.; Writing—original draft, S.L. and J.Z.; Writing—review & editing, Z.L. All authors have read and agreed to the published version of the manuscript.

**Funding:** This work was partially supported by the National Natural Science Foundation of China (61703135) and the Program of Introducing the Foreign Talent of Hebei Province.

**Institutional Review Board Statement:** Not applicable. The study did not involve humans or animals.

**Informed Consent Statement:** Not applicable. The study did not involving humans.

**Data Availability Statement:** The data presented in this study are available on request from the corresponding author. The data are not publicly available due to privacy.

**Conflicts of Interest:** The authors declare no conflict of interest.

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
