# Peer review of "An Experience Transfer Approach for the Initial Data of Iterative Learning Control"

_applsci, doi:10.3390/app11041631_

Round 1

Reviewer 1 Report

Article: An Iterative Learning Control Strategy with 2 Experience Transfer
Authors: Shaozhe Liu, Zuojun Liu, Jie Zhang and Dong Hu

This paper addresses an iterative learning control (ILC) using experience data obtained from original ILC. A new ILC  strategy with experience transfer is proposed according to the relation between the new and the original systems.

The iterative control scheme has been already developed. As far as the reviewer knows, the contribution of the paper is not enough to be published in a journal paper.
The usage of experience data is only the change of input data for the original ILC. The analysis and the new method for ILC has not been addressed.
If one uses more data, clearly it shows better performance. I do not want to reccommend the current paper. The convergence result is not also a new one.

Author Response

Response to Reviewer 1 Comments

Comment 1-1: The iterative control scheme has been already developed. As far as the reviewer knows, the contribution of the paper is not enough to be published in a journal paper. The usage of experience data is only the change of input data for the original ILC. The analysis and the new method for ILC has not been addressed. If one uses more data, clearly it shows better performance. I do not want to recommend the current paper. The convergence result is not also a new one.

Author response: We really appreciate your efforts and comments on our manuscript. We have revised our manuscript according to your comments and suggestions.

  • Regarding to the comments “The iterative control scheme has been already developed.”and “the contribution of the paper is not enough”, we think it might be a misleading caused by the title of our paper. The former title “An Iterative Learning Control Strategy with Experience Transfer” gives reviewer an impression that a new ILC strategy will be presented in the paper. However, our work is mainly about the initial learning data of ILC, not a new ILC strategy. To avoid the misunderstanding, we change the paper title into “An Experience Transfer Approach for the Initial Data of Iterative Learning Control”.Besides, as the abstract and introduction are also not well stated, we rewrite the abstract and the 6th and 7th paragraph in the Section “Introduction”.
  • The comment “The usage of experience data is only the change of input data for the original ILC”, is true. However, the way of usage proposed in our paper is new. The experience data are not directly used, but in reference with the relation of new and original system, which are called “experience inheritance”and “experience transform”. It is a new approach that no similar papers are found.
  • Regarding to the comments “The convergence result is not also a new one”, as our approach is used for the ILC system with parameter changes, so when there is change in B or C, the learning gain should be regulated accordingly to guarantee the convergence condition and speed. We add some extra illustration to explain it in Section 6.

We updated the manuscript by adding the following parts with yellow high lighting indicating changes.

  • The paper title
  • The abstract
  • The 6th and 7th paragraph in the Section 1 “Introduction” (lines 91-101)
  • The 1st paragraph in the Section 6 “Convergence Analyze” (lines 327-332)

We hope the update may state our contribution clearly. If there is more suggestion, we will be appreciated and try our best to revise our paper.

Reviewer 2 Report

This paper focuses on ILC with transfer of expertise through recombination, translation, and amplitude leading to adaptation from the original ILC for transfer to the new ILC's initial control data. It seems that simulation results are presented. In my opinion, the presentation of both the equations and some of the results of this paper is not entirely clear. My detailed comments are as follows: -

  • For the abstract, it should be rewritten more precisely to include the main ideas and actual contributions of the manuscript. The abstract contains only general ideas about the ILC and does not fully cover what was accomplished in the manuscript. This should be presented in an interesting way. For example, there is a re-idea of the usual iterative learning control in its known form. I propose to include the main problem, clarify the type of iterative learning control adopted in the manuscript, and add the number of iterations achieved after adopting the new approach.

  • There is many typos and format problems. Such as Line 31, 32, 34, 36, 39, 41 on Page 1, the first line, 44 on page 2. In addition, in page 2 from line (45 to 47), I don’t understand what you mean by "without involving the thoughts of making effective use of the previous control experience data" when the desired trajectory changes. This paragraph needs more clarity about the " making effective use of the previous control".

  • The applications of ILC to some complicated systems should be mentioned in Introduction, such as,

1- " Application of Norm Optimal Iterative Learning Control to Quadrotor Unmanned Aerial Vehicle for Monitoring Overhead Power System"

2- "Distributed control of multiple flexible manipulators with unknown disturbances and dead-zone input".

Both papers have the basis of new practical application and are used for more than 4 degree of freedom. 

  • For the reference 21, it is preferable to replace it because it is an old research and the results presented in the article are unrealistic, so this article should be added, for example, "An Upper Limit for Iterative Learning Control Initial Input Construction Using Singular Values". This paper has new novel such as an upper limit to the initial choice construction for the input signal for trial one is set such that the system would not tend to respond aggressively due to the uncertainty.

  • The type of ILC algorithm is simple structure P in discrete-time, I need to know did you have any limitations in Simulation test application, such as unguaranteed monotonic convergence and increases sensitivity to noise??In this case, try to mention in the conclusion, or it may be the method (transfer of experience) that you used is able to solve this problem??

  • Can you explain why you taken a random estimation deviation in simulation test and is this happen with the original system and new system?

  • In section 6 for the convergence analyse, the author claim that the iteration turns could decrease significantly new system comparing to original system, but what's clear from the simulation test that almost both results reach same error value with about 17 or 18 iterations. I think you need to add table to show the reduction of error vs the iteration number for both systems new and old. In addition, regarding to the line 291 in page 10 "so the iteration turns needed for the given convergent accuracy could decrease significantly" This sentence does not clearly show how this happens, so the author may need to use some keywords such as accelerated reach, error reduction, and Monotonic convergence that can explain the outcome he guaranteed.

Best wishes

Author Response

Response to Reviewer 4 Comments

Comment 2-1: For the abstract, it should be rewritten more precisely to include the main ideas and actual contributions of the manuscript. The abstract contains only general ideas about the ILC and does not fully cover what was accomplished in the manuscript. This should be presented in an interesting way. For example, there is a re-idea of the usual iterative learning control in its known form. I propose to include the main problem, clarify the type of iterative learning control adopted in the manuscript, and add the number of iterations achieved after adopting the new approach.

Author response: We really appreciate your efforts and comments on our manuscript. We have revised our abstract according to your comments and suggestions.

We updated the abstract with yellow high lighting indicating changes. The sentence of re-idea of the usual iterative learning control is erased. Our contribution on experience transfer for the initial data in the ILC of new system is directly presented in the abstract. The main problem, the approach and the number of iterations achieved after adopting the new approach are all added.

We hope the update may state our main ideas and actual contributions precisely.

Comment 2-2: There are many typos and format problems. Such as Line 31, 32, 34, 36, 39, 41 on Page 1, the first line, 44 on page 2. In addition, in page 2 from line (45 to 47), I don’t understand what you mean by "without involving the thoughts of making effective use of the previous control experience data" when the desired trajectory changes. This paragraph needs more clarity about the "making effective use of the previous control".

Author response: We are so apologize for our poor writing skills. We correct the typos and format mistakes pointed by the reviewer, as well as some other mistakes found by ourselves. Most format problems marked with green wavy line by WORD are due to the omitting of blank between the last word and the citation mark of references. We have corrected these mistakes.

Besides, some vague and “Chinese style English” sentences are rewritten. The updated words and sentence are marked with yellow high lighting to indicate the changes. (lines 54-56, 85-89, 91-101 etc.)

Comment 2-3: The applications of ILC to some complicated systems should be mentioned in Introduction, such as,

1- " Application of Norm Optimal Iterative Learning Control to Quadrotor Unmanned Aerial Vehicle for Monitoring Overhead Power System"

2- "Distributed control of multiple flexible manipulators with unknown disturbances and dead-zone input".

Both papers have the basis of new practical application and are used for more than 4 degree of freedom.

Author response: We really appreciate the papers you recommend. Both papers present feasible approaches for the improvement of ILC. These two papers are referred in lines 50-54, listed in the reference 13 and 14.

The added paper review is marked with yellow high lighting in the first paragraph of page 2.

Comment 2-4: For the reference 21, it is preferable to replace it because it is an old research and the results presented in the article are unrealistic, so this article should be added, for example, "An Upper Limit for Iterative Learning Control Initial Input Construction Using Singular Values". This paper has new novel such as an upper limit to the initial choice construction for the input signal for trial one is set such that the system would not tend to respond aggressively due to the uncertainty.

Author response: We appreciate the reviewer’s suggestion. The former reference 21 is really old. We replace it with the recommended paper, as well as the corresponding introduction in the literature review in lines 83-85.

The review of new paper is marked with yellow high lighting in the second paragraph of page 2.

Comment 2-5: The type of ILC algorithm is simple structure P in discrete-time, I need to know did you have any limitations in simulation test application, such as unguaranteed monotonic convergence and increases sensitivity to noise.

In this case, try to mention in the conclusion, or it may be the method (transfer of experience) that you used is able to solve this problem?

Author response: Our paper is mainly about the initial learning data of ILC, not a new ILC strategy. The performances of monotonic convergence and sensitivity to noise are not involved in our paper. The main contribution is to reduce the convergence iterations of new ILC process when the system parameter changes. There isn’t any limitation in simulation test application. The P-ILC is used just as an example. PD-ILC, D-ILC or other ILC can also be used for simulation.

To make it clearly, we accept your suggestion and mention it in the conclusion, as shown in final paragraph of paper with the yellow high lighting. (lines 348-351)

Comment 2-6: Can you explain why you taken a random estimation deviation in simulation test and is this happen with the original system and new system?

Author response: The experience transfer approach needs the relation/difference information of original system and new system. The experience of former ILC will be transformed according to this information. To prove the effect of approach, some random estimation deviation is added on this information. The simulation results show that the approach is effective even though the information is not accurately measured. The inaccuracy measurement happens both with the original system and new system. To make the meaning clear, we change “random estimation deviation” into “random measurement error”.

The above explain is added in lines 267-271, with the yellow high lighting.

Comment 2-7: In section 6 for the convergence analyze, the author claim that the iteration turns could decrease significantly new system comparing to original system, but what's clear from the simulation test that almost both results reach same error value with about 17 or 18 iterations. I think you need to add table to show the reduction of error vs the iteration number for both systems new and old. In addition, regarding to the line 291 in page 10 "so the iteration turns needed for the given convergent accuracy could decrease significantly" This sentence does not clearly show how this happens, so the author may need to use some keywords such as accelerated reach, error reduction, and Monotonic convergence that can explain the outcome he guaranteed

Author response: We appreciate the reviewer’s comments and suggestion.

  • Regarding to “both results reach same error value with about 17 or 18 iterations”, we check the simulation program, and find that it is caused by the random measurement error. The random measurement error should be a random value in the first iteration but fixed in the following to representthe measure error. But in our program, it changes in every iteration and causes the incorrect convergence result. We have corrected it in the manuscript, as shown in Fig.8.
  • We add 3 tables according to your suggestion to show the reduction of error vs the iteration number for both systems new and old. (lines 240, 271, 316)
  • Regarding to the vague sentence of line 291 in page 10, we change it “so the system could accelerated reach the required convergent accuracy with less error reduction iteration turns” according to your suggestion. (in this manuscript, it is lines 338-340 )

All the updates are marked with the yellow high lighting in the manuscript.

If there is more suggestion, we will be appreciated and try our best to revise our paper.

Reviewer 3 Report

The authors are asked to make the text more comprehensible. Thus, the text of lines 111-112 should be clarified. 

Also, the structure of equations (3), (4) should be made explicit, in the sense of clearly defining the variables, parameters and the connection between the time domain and the frequency domain. The asterisk as a substitute for the multiplication operation may well be missing.

In relations (5), (6), the variable z must be guessed by the reader, because in the text it is not explained at all. Relationships (5), (6) should be fully clarified.

In fact, the authors' style seems inclined towards hermeticism, and not towards clarity: how were the relations (7) deduced, how were relations (8) - (10) transcribed?

Author Response

Response to Reviewer 3 Comments

Comment 3-1: The authors are asked to make the text more comprehensible. Thus, the text of lines 111-112 should be clarified.

Author response: We really appreciate your efforts and comments on our manuscript. Some “Chinese-style English” or vague sentences are revised in our manuscript, including the text of lines 111-112 (now, it is the text of lines 133-138 in the updated manuscript due to the other revise in the lines before it.).

All the updates are marked with the yellow high lighting in the manuscript.

Comment 3-2: Also, the structure of equations (3), (4) should be made explicit, in the sense of clearly defining the variables, parameters and the connection between the time domain and the frequency domain. The asterisk as a substitute for the multiplication operation may well be missing.

Author response: The definition of the variables, parameters and the explicit about the connection between the time domain and the frequency domain are added in the lines 128-133. The asterisk in the equation is removed. The updated part is marked with the yellow high lighting in the manuscript.

Comment 3-3: In relations (5), (6), the variable z must be guessed by the reader, because in the text it is not explained at all. Relationships (5), (6) should be fully clarified.

Author response: We add the illustration for variable z in the lines 200-201, which is “z-i denotes the delayed effect of pulse input in the discrete system”.

The relationship between equation (5) and (6) is explained in the lines 209-212, which is “In equation (5), the output response sequences ya of control data sequence ua combine together to form the whole desired trajectory. So do those of yb. The control data sequences in the equation (6) are desired to have the same effect as that of equation (5). The relation can be obtained via the discrete impulse response of two systems shown in Fig.3”.

Both the two updated parts are marked with the yellow high lighting in the manuscript.

Comment 3-4: In fact, the authors' style seems inclined towards hermeticism, and not towards clarity: how were the relations (7) deduced, how were relations (8) - (10) transcribed?

Author response: Relations (7) is based on the equation (5) and (6), and is explained in lines 209-212 together with comment 3-3. In lines 224-230, the transcribing process of relations (8) - (10) are explained. We hope the update may state our idea clear.

The updated part is marked with the yellow high lighting in the manuscript.

Reviewer 4 Report

Some comments, suggestions:

  • This paper proposes to use the improved Iterative Learning Control (ILC) of recombining, translational and amplitude adjusting respectively. Based on the presented simulation tests, it can be concluded that improved ILC forms the ideal initial iteration learning data with relatively small initial error. Unfortunately, the authors presented too few simulation tests. If, in addition, a different range of simulation tests, that confirm the effectiveness of the improved ILC were carried out, I propose to complete the article with additional results and additional conclusions
  • The authors have inserted the same figures in two places in the article - please explain:

        Figure 2. Step Response of Systems with Different Time Constants.

        Figure 7. Amplitude Difference of Step Response.

  • Improve the quality of the attached Figure 4. ILC of Recombined Experience Transform.

Author Response

Response to Reviewer 4 Comments

Comment 4-1: This paper proposes to use the improved Iterative Learning Control (ILC) of recombining, translational and amplitude adjusting respectively. Based on the presented simulation tests, it can be concluded that improved ILC forms the ideal initial iteration learning data with relatively small initial error. Unfortunately, the authors presented too few simulation tests. If, in addition, a different range of simulation tests, that confirm the effectiveness of the improved ILC were carried out, I propose to complete the article with additional results and additional conclusions

Author response: We really appreciate your careful work on our manuscript. Your comments are of great value for us. We add a series of new simulation in Section 7 Conclusion.

It may be a little weird to insert some figures in “Section 7 Conclusion”. The reason is below.

On 8 Jan, we received editor’s e-mail, in which we are informed to revise the manuscript according to the comments from 3 reviewers and return the updated manuscript no later than 18 Jan. We downloaded the comments of 3 reviewers and begin to revise our paper. Yesterday, 17 Jan, we finished the revising and begin to upload the updated manuscript. At that time, we found your on-line comments of 9 Jan, the comment from the 4th reviewer. As there is almost no time before the deadline, it is hard to add another simulation in Section 3, 4 and 5. If so, the structure of paper should be re-arranged. Besides, more sentences, equations and figures are also need be inserted. In so short time, we can only add it in the final section. If there is more time, we would make it in a better way.

Please understand our difficulty.

Comment 4-2:The authors have inserted the same figures in two places in the article - please explain:

        Figure 2. Step Response of Systems with Different Time Constants.

        Figure 7. Amplitude Difference of Step Response.

Author response: It is really a terrible mistake due to our careless work. We are so sorry about it. And thank you very much for your careful review. We change the wrong Figure.2 and replace it with the correct one.

Comment 4-3: Improve the quality of the attached Figure 4. ILC of Recombined Experience Transform.

Author response: A clear figure 4 is redrawn and inserted.

If there is more suggestion, we will be appreciated and try our best to revise our paper.

Round 2

Reviewer 2 Report

The issues highlighted in the previous revision step have been considered.

No more comments. It may be accepted now.